# Novel UHRF1-MYC Axis in Acute Lymphoblastic Leukemia

**DOI:** 10.3390/cancers14174262

**Published:** 2022-08-31

**Authors:** Soyoung Park, Ali H. Abdel Sater, Johannes F. Fahrmann, Ehsan Irajizad, Yining Cai, Hiroyuki Katayama, Jody Vykoukal, Makoto Kobayashi, Jennifer B. Dennison, Guillermo Garcia-Manero, Charles G. Mullighan, Zhaohui Gu, Marina Konopleva, Samir Hanash

**Affiliations:** 1Department of Clinical Cancer Prevention, The University of Texas MD Anderson Cancer Center, Houston, TX 77030, USA; 2Department of Biostatistics, The University of Texas MD Anderson Cancer Center, Houston, TX 77030, USA; 3Department of Basic Pathology, School of Medicine, Fukushima Medical University, Fukushima 960-1295, Japan; 4Department of Leukemia, The University of Texas MD Anderson Cancer Center, Houston, TX 77030, USA; 5Department of Pathology, St. Jude Children’s Research Hospital, Memphis, TN 38105, USA

**Keywords:** acute lymphocytic leukemia, UHRF1, c-Myc

## Abstract

**Simple Summary:**

We provided evidence that ubiquitin-like, containing PHD and RING finger domain, 1 (UHRF1) is overexpressed in acute lymphocytic leukemia (ALL). We further showed that UHRF1 directly interacts and regulates c-Myc expression to enable ALL cell growth through the cMYC-CDK4/6 phosphoRb-signaling axis.

**Abstract:**

Ubiquitin-like, containing PHD and RING finger domain, (UHRF) family members are overexpressed putative oncogenes in several cancer types. We evaluated the protein abundance of UHRF family members in acute leukemia. A marked overexpression of UHRF1 protein was observed in ALL compared with AML. An analysis of human leukemia transcriptomic datasets revealed concordant overexpression of UHRF1 in B-Cell and T-Cell ALL compared with CLL, AML, and CML. In-vitro studies demonstrated reduced cell viability with siRNA-mediated knockdown of UHRF1 in both B-ALL and T-ALL, associated with reduced c-Myc protein expression. Mechanistic studies indicated that UHRF1 directly interacts with c-Myc, enabling ALL expansion via the CDK4/6-phosphoRb axis. Our findings highlight a previously unknown role of UHRF1 in regulating c-Myc protein expression and implicate UHRF1 as a potential therapeutic target in ALL.

## 1. Introduction

UHRF family proteins UHRF1 and UHRF2 are characterized by functional domains consisting of ubiquitin-like, PHD (plant homeodomain), RING (really interesting new gene), and methyl-DNA-binding SRA (SET and RING-associated) motifs [1,2]. UHRF1 expression in epithelial tumors positively correlates with the stage, grade, drug resistance, and maintenance of cancer stem cell characteristics and has been suggested as a putative oncogene [3,4,5,6,7,8,9]. Moreover, high tumoral expression of UHRF1 is associated with poor prognosis of several cancer types, including hepatocellular carcinoma, pancreatic, breast, and bladder cancer [10,11,12,13].

UHRF1 plays a role in cell cycle progression and cell growth in proliferating adult cells and during embryogenic development [14,15,16]. Nuclear-localized UHRF1 has been shown to function in regulating and preserving epigenetic DNA methylation throughout the cell cycle [1,2]. Specifically, during DNA replication, UHRF1 methylates several substrates, including CpG dinucleotides, histone H3, and p53, resulting in altered chromatin structure and protein function [17,18,19]. UHRF1 overexpression has been shown to drive DNA hypomethylation by reducing DNA methyltransferase 1 (DNMT1) levels and limiting its access to hemimethylated DNA, resulting in genomic instability that, when coupled with TP53 inactivation and senescence bypass mechanism(s), promotes tumorigenesis [20]. Loss of UHRF1 has also been reported to promote cell cycle arrest at the G1/S phase transition through the activation of the p53/p21^cip1/waf1^-dependent DNA damage response [21,22]. In another study, the knockdown of UHRF1 in HCT116 cancer cells induced cell cycle arrest in G2/M and promoted caspase-8 dependent apoptosis [23].

UHRF2 shares high structural homology with UHRF1. However, UHRF2 is not functionally redundant in maintaining DNA methylation [24,25] and, unlike UHRF1, the expression of UHRF2 in human cancers tends to be low due to mutation [26], copy number loss [27], or CpG hypermethylation of the promoter region [28]. The specific activities of UHRF proteins are likely diverse, given their structural complexity, and the roles of UHRFs outside of DNA methylation, particularly in the context of cancer, need to be further explored.

In our study, we investigated the expression of UHRF1 in leukemias, which revealed the enrichment of UHRF1 in T-cell and B-cell ALL compared with CLL, AML, and CML. Functional studies revealed that UHRF1 contributes to ALL expansion via a mechanism that includes upregulated c-Myc protein expression.

## 2. Materials and Methods

### 2.1. Patient Samples

Leukemia samples for protein analysis were obtained from the MD Anderson Cancer Center following IRB approval. Additional leukemia data were obtained through the St. Jude’s Children’s Research Hospital; Children’s Oncology Group (COG); and the Therapeutically Applicable Research to Generate Effective Treatments (TARGET) project. Institutional review boards from the following institutions were responsible for oversight: Ann & Robert H. Lurie Children’s Hospital, Fred Hutchinson Cancer Research Center, National Cancer Institute, St Jude’s Children’s Research Hospital, The Children’s Hospital of Philadelphia, The University of New Mexico, Texas Children’s Hospital, and The Hospital for Sick Children. Informed consent was obtained from all subjects.

### 2.2. Gene Expression Datasets

Gene expression data for UHRF1, MYCN, MYCL1, MYC, CDK6, and CDK4 in the Haferlach [29] and Gu [30] leukemia datasets were downloaded from the Oncomine database [31] and the St. Jude Cloud Interactive Visualization Portal (https://pecan.stjude.cloud/proteinpaint/study/PanALL (accessed on 8 October 2020)) [30]. Transcriptomic profiling data for the Haferlach leukemia dataset were generated using the Human Genome U133 Plus 2.0 Array [29]. For the Gu leukemia dataset, RNA-seq was performed using TruSeq library preparation and HiSeq 2000 and 2500 sequencers (Illumina, San Diego, CA, USA) [30].

### 2.3. Cell Culture

All leukemia cell lines were maintained in RPMI1640 with 10% fetal bovine serum (FBS). Cells were maintained at 37 °C in a humidified atmosphere with 5% CO_2_ and were verified negative for mycoplasma contamination before experiments.

### 2.4. RNA Interference-Mediated UHRF1 Knockdown

UHRF1 siRNA 1 (s26553; Ambion, Life Technologies, Austin, TX, USA) and UHRF1 siRNA 2 (s26555; Ambion, Life Technologies) were purchased. Negative control siRNA was purchased from Life Technologies (4390849). For the transfection experiments, the cells were seeded at a cell density of 2 × 10^6^/well in 24-well plates. siRNAs (400 nM) were transfected using a Neon Transfection Electroporation system (Invitrogen, Waltham, MA, USA) according to the manufacturer’s protocol. Twenty-four and forty-eight h after the transfection, the cells were collected for RNA and protein isolation and subjected to RT-qPCT and Western blot analysis, respectively.

### 2.5. Total RNA Isolation and Quantitative Real-Time PCR Analysis

The total RNA was extracted using RNeasy Mini Kit (Qiagen, Hilden, Germany) according to the manufacturer’s protocol. For the RT-qPCR analysis, 2 µg of total RNA was reverse-transcribed into cDNA using the High-Capacity cDNA Archive Kit (Applied Biosystems, Waltham, MA, USA). The gene expression levels of UHRF1 and 18S (loading control) were assessed using a Real-time PCR System (Bio-Rad) with Tagman Master Mix II (4440040; Thermo Scientific, Waltham, MA, USA). TaqMan probes and primers were purchased for UHRF1 (product ID: Hs 01086727_m1; Thermo Fisher Scientific, Waltham, MA, USA), MYC (product ID: Hs00153408_m1; Thermo Fisher Scientific), and 18S (Hs99999901_s1; Thermo Fisher Scientific). The expressions of UHRF1 and MYC mRNA relative to 18S mRNA were determined using the 2^−ΔΔCT^ method. RT-PCR analyses were done in triplicate.

### 2.6. Cell Proliferation Assay

MOLT4 and REH cells were seeded in 96-well plates at a density of 1 × 10^5^ cells/well. Cell viability was determined using the CellTiter 96 Aqueous One Solution Cell Proliferation Assay kit (Promega, Madison, WI, USA) after 24 and 48 h post-transfection. All experiments were performed in biological triplicates.

### 2.7. Immunoblotting for siRNA-Mediated Knockdown in T-ALL and B-ALL Cells

The total cell lysate protein was extracted using RIPA Buffer (Invitrogen). The protein concentration was measured with a Bicinchoninic Acid (BCA) protein assay reagent (Pierce), and 30 μg of protein extract was loaded onto 4–15% SDS-PAGE gradient gels from Bio-Rad. The proteins in the gels were transferred onto the PVDF membrane. After blocking with TBST containing 5% non-fat dry milk, the membranes were blotted with antibodies against UHRF1 (sc-373750, Santa Cruz, Dallas, TX, USA), MYC (5605S, Cell Signaling, Danvers, MA, USA), CDK4/CDK6 (ab108357, Abcam (Cambridge, UK) /sc-7961, Santa Cruz), Phospho-Rb (Ser807/811) (8516, Cell Signaling), RB (9309, Cell Signaling), PAPR (9542S, Cell Signaling), Cleaved Caspase 3 (Asp175) (9661S, Cell Signaling), GAPDH (ab9485, Abcam), and Tubulin (2128, Cell Signaling). The protein signals were detected with an ECL system (Bio-Rad) and with X-ray film.

### 2.8. Cell Cycle Analysis

The REH and MOLT4 ALL cells were collected 72 h after transfection for a cell cycle analysis via flow cytometry. For the cell cycle analysis, the cells were washed three times with cold phosphate buffer saline (PBS), fixed with 70% ethanol at −20 °C overnight, washed with PBS, re-suspended in 0.5 ml of propidium iodide staining solution (PI, Invitrogen; Thermo Fisher Scientific, Inc.), and incubated for 30 min in the dark at room temperature. The cells were subsequently analyzed on a Gallios Flow Cytometry System for cell cycle distribution analysis.

### 2.9. Immunoprecipitation

The cells were lysed in IP Lysis buffer (87788, Thermo Scientific) along with a protease inhibitor cocktail (Roche Applied Science, Upper Bavaria, Germany) at 4 °C for 30 min. Thereafter, the cell extract was centrifuged at 14,000 rpm for 15 min at 4 °C, and 5% of the cell extract was kept for input. To conjugate the primary antibody, the cell extracts were incubated with 2 µg of the anti-UHRF1 (Santa Cruz), anti-MYC (Cell Signaling), or rabbit IgG antibody and Dynabeads protein G (Thermo Fisher Scientific) for overnight incubation at 4 °C. The antibody–antigen complex was washed three times with PBS. Laemmli buffer (Bio-Rad, Hercules, CA, USA) was subsequently added to elute the precipitated proteins, followed by Western blotting using anti-UHRF1 (sc-373750, Santa Cruz) and anti-c-Myc antibodies (5605S, Cell Signaling). The immunoblot conditions were the same as described for the siRNA-treated samples.

### 2.10. Mass Spectrometry Analysis

After Co-IP, anti-UHRF1 and anti-IgG IP samples from REH were electrophoresed in 4–15% SDS-PAGE gels (Bio-Rad, Hercules, CA, USA) and stained with the Zinc Reversible Stain kit (Thermos Scientific). The band areas corresponding to UHRF1 (90 kDa) and cMyc (50 kDa) were excised from the gel, placed into a 1.5 mL tube, and in-gel digested with trypsin [32]. The tryptic peptides extracted from the bands were analyzed using an Orbitrap ELITE Mass Spectrometer (Thermo Scientific) and searched with Proteome Discoverer 1.4 (Thermo Scientific). Sequest HT was used as a search engine, with the parameters including fixed modification of Cys alkylated with diethylcarbamidomethyl (+113.084) and variable modification of Met oxidation (+15.995). A mass error of 10 ppm was allowed for parent MS1, and 0.5 Da was allowed for the MS2 fragments. The data were searched against the Uniprot Human database, 2017, and we further filtered the data with FDR = 0.01.

### 2.11. Proteomic Analysis following UHRF1 siRNA-Mediated Knockdown

For the proteomic analysis of UHRF1, whole cell lysates of ALL and AML cells obtained from patients were lysed in PBS containing octyl-glucoside (1% *w*/*v*) and protease inhibitors (complete protease inhibitor cocktail, RocheDiagnostics, Basel, Switzerland), followed by sonication and centrifugation at 20,000× *g* with a collection of the supernatant and then filtration through a 0.22 µm filter. Two milligrams of whole cell extracts (WCE) proteins were reduced in DTT and alkylated with acrylamide before fractionation by RP-HPLC. A total of 24 fractions were analyzed by LC-MS/MS per the cell line. The acquired data were processed and searched against the Uniprot proteome database through ProteinLynx Global Server (PLGS, Waters Company, Milford, MA, USA), with a false discovery rate of 4%. We analyzed the proteins in the MOLT4 (T-ALL) cells after the UHRF1 knockdown in comparison with the control using tandem mass tag (TMT) LC-MS/MS, as previously described [33,34,35]. A total of 100 µg of protein was used for TMT-labeling per channel. The siNC was labeled with TMT126, siUHRF1-1 was labeled with TMT127, and siUHRF1-2 was labeled with TMT128, followed by quenching with hydroxylamine and drying using a SpeedVac. The TMT labeled peptide mixture was subsequently fractionated into ten fractions for LC/MS analysis using a Q Exactive Mass Spectrometer (Thermo Fisher). The acquired LC-MS/MS data were processed by the Proteome Discoverer 1.4 (Thermo Scientific). Sequest HT was used as a search engine, with the parameters including fixed modification of Cys alkylated with diethylcarbamidomethyl (+113.084), Lys with TMT (+229.163, N-terminal and Lys) and variable modification of Met oxidation (+15.995). A mass error of 10 ppm was allowed for parent MS1, and 0.02 Da was allowed for the MS2 fragments. The data were searched against the Uniprot Human database, 2017, and further filtered with FDR = 0.01, and the TMT ratios were quantified.

### 2.12. Statistical Analysis

For two-class comparisons, the statistical significance was determined using the Wilcoxon rank sum test unless specified. Statistical significance was considered to be *p*-values < 0.05. For more than two-class comparisons, we reported the adjusted *p*-values of Dunn’s multiple comparison test. The results are presented as the means ± standard deviation (mean ± SD). Ingenuity Pathway Analyses (Version 49309495, Qiagen, Hilden, Germany) were performed using proteins that were differentially expressed (fold change ≥ 1.2 or ≤0.83), following a siRNA-mediated knockdown of UHRF1. The statistical significance was determined by Fisher’s exact test. A false-discovery rate adjustment was performed using the Benjamini–Hochberg (BH) method.

## 3. Results

### 3.1. UHRF1 Is Overexpressed in ALL

An initial analysis of UHRF1 protein expression was performed using leukemia cells from subjects with Philadelphia-positive (Ph+) or Philadelphia-like (Ph-like) ALL (*n* = 56 subjects) and with AML (*n* = 76 subjects). Statistically significantly higher levels of UHRF1 protein were observed in Ph+/Ph-like ALL compared with those of AML (Wilcoxon rank-sum test *p*: 0.0001) (Figure 1A). To further determine the expression of UHRF1 in ALL subtypes, we compared the UHRF1 gene expression in the Haferlach leukemia dataset [29], which revealed that the UHRF1 mRNA expression was statistically significantly higher (Dunn’s multiple comparison test, adjusted *p* < 0.0001) in all ALL subsets (B-ALL, Pre-B-ALL, and T-ALL) compared with those of CLL, CML, and AML (Figure 1B; Table A1 in the Appendix B). A comparison of UHRF1 mRNA expression among B-ALL subtypes in the Gu leukemia dataset [30] revealed the UHRF1 gene expression to be highest in the ETV6-RUNX1 subtype (Figure A1 in the Appendix B). A protein-level analysis of UHRF1 in healthy-donor bone marrow-derived cells and human ALL. Through Western blot, the AML cell lines confirmed UHRF1 expression to be highly elevated in ALL with levels in AML and below detection in healthy-donor bone marrow cells (Figure A2 in the Appendix B). Given these findings, we focused our subsequent analyses toward elucidating the biological role of UHRF1 in the context of ALL.

### 3.2. UHRF1 Regulates the c-Myc-CDK4/6-pRB Axis in ALL

To determine the biological relevance of UHRF1, we performed a siRNA-mediated knockdown of UHRF1 in the T-ALL cell line MOLT4 and B-ALL cell line REH, the results of which revealed a statistically significant reduction in cell viability following the loss of UHRF1 expression in both cell lines (Figure 1C; Figure A3 in the Appendix B).

To elucidate the mechanism by which UHRF1 affects ALL cell viability, we performed quantitative proteomics in the T-ALL MOLT4 cell line following a siRNA-mediated knockdown of UHRF1. A total of 1166 proteins were quantified by LC/MS analysis, of which 179 proteins were elevated (fold change ≥ 1.2), and 166 proteins were reduced (Fold change ≤ 0.83). An Ingenuity Pathway Analysis (IPA) applied to these 345 differentially expressed proteins identified c-Myc as the top upstream transcription factor predicted to be inhibited (BH-adjusted two-sided *p*-value: 1.74 × 10^−25^). EIF2 signaling was identified as the top downregulated canonical pathway (two-sided *p*-value: 5.01 × 10^−21^) (Figure 2A; Table A2 in the Appendix B). Consistent with the inhibition of a c-Myc-related pathway, our proteomic analyses revealed the reduced protein expression of several annotated c-Myc-downstream targets, including CDK6, an integral cyclin-dependent kinase that promotes cell cycle progression [36], following a siRNA-mediated knockdown of UHRF1 (Figure 2B).

The c-Myc is the upstream transcriptional regulator of the cyclin-dependent kinases CDK4 and CDK6 [37,38]. The CDK4/6 complex promotes cell cycle transition from the G1 phase to the S phase via phosphorylation of Rb and activation of E2F [39]. Spearman correlation analyses between UHRF1 mRNA levels and the gene expression of MYC family members (MYC, MYCN, and MYCL1), as well as CDK4 and CDK6 in the Haferlach and Gu leukemia datasets [29,30], indicated positive associations between UHRF1 and c-Myc, CDK4, and CDK6 in ALL (Figure 2C,D).

To evaluate the extent to which UHRF1 regulates the c-Myc-CDK4/6-phosphoRb axis in ALL, we quantified the expression of c-Myc, CDK4, and CDK6 following a siRNA-mediated knockdown of *UHRF1* in T-ALL cell lines MOLT4 and PF832 and B-ALL cell lines BALL1 and REH (Figure A3A in the Appendix B). The knockdown of UHRF1 resulted in a pronounced decrease in c-Myc protein expression in all four ALL cell lines. Minimal changes in c-MYC mRNA levels were observed following a siRNA-mediated knockdown of UHRF1. However, these were not concomitant with changes observed at the protein level (Figure 3A,B; Figure A3A in the Appendix B). A loss of c-Myc protein expression was associated with concordant decreases in the CDK4 and CDK6 protein expressions and a decrease in Rb phosphorylation (Figure 3A,B). Notably, the siRNA-mediated knockdown of Myc had minimal impact on the UHRF1 mRNA levels (Figure A3B in the Appendix B). Cell cycle analyses following the siRNA-mediated knockdown of UHRF1 in REH and MOLT4 ALL cells resulted in a modest accumulation of cells in the G0/G1 phase. These changes were met with increases in the protein expression of cleaved caspase-3 and cleaved PARP (Figure 4A,B), indicating an induction of apoptosis. These findings implicate UHRF1 in regulating the c-Myc-CDK4/6-phoshoRb axis, thereby providing a biological basis for our observation of reduced ALL viability following a knockdown of UHRF1 (Figure 1C).

### 3.3. UHRF1 Promotes the Regulation of c-Myc Protein

The regulation of the c-Myc protein expression is controlled through complex systems, including targeted degradation by the ubiquitin-proteasome system [40]. UHRF1 is known to exert ubiquitin ligase functions [41]. We assessed whether endogenously expressed UHRF1 directly interacted with c-Myc using co-immunoprecipitation (co-IP) assays applied to MOLT4 T-ALL cells and REH B-ALL cells. Western blotting of the IP showed that UHRF1 and c-Myc were enriched in the pulldown products from the two cell lines (Figure 5A,B). To further confirm the results of co-IP using the Western blotting results, anti-UHRF1 IP and control anti-IgG IP from REH cells (Figure 5B) were loaded onto SDS-PAGE, followed by Zn staining. The protein bands corresponding to UHRF1 and c-Myc based on the Western blot image were cut, in-gel digested, and analyzed by LC-MS/MS. The UHRF1 unique peptides and c-Myc unique peptides were identified in the anti-UHRF1 IP (FDR = 1%) but not in the control anti-IgG IP (Table A3 and Figure A4 in the Appendix B).

## 4. Discussion

In our study, we investigated the expression levels of UHRF1 in ALL and found that UHRF1 was statistically significantly elevated in ALL compared with its level in other types of leukemia, both at the protein and gene expression levels. We further showed that UHRF1 directly interacts with c-MYC and that knockdown of UHRF1 in T-ALL and B-ALL cells reduces cell viability via inhibition of the c-Myc-CDK4/6-phosphoRb axis and induction of apoptosis.

Aberrations in UHRF1 expression were linked with aggressiveness in several cancer types, including ALL [42,43]. To date, the oncogenic role(s) of UHRF1 have largely been described in the context of epigenetic regulation through UHRF1-mediated DNA hypermethylation and histone modification. For instance, the upregulation of UHRF1 in breast cancer was shown to result in hypermethylation and inhibition of *BRCA1* by forming an inhibitory transcriptional complex consisting of HDAC, DNMT1, and G9a over its promotor [44]. In colorectal cancer, UHRF1-mediated methylation of a peroxisome proliferator-activated receptor (PPARγ) was shown to be a key determinant of disease progression [45]. Similarly, a promoter hypermethylation of G-protein signaling 2 (RGS2) gene by UHRF1 resulted in gene suppression and enhanced carcinogenesis in bladder cancer [46]. An increased cellular proliferation of endometrial cancer was shown to be linked to UHRF1-mediated H3R8 di-methylation of the *SOCS3* and *3OST2* promoter [47]. UHRF1 was also shown to epigenetically repress several tumor suppressor genes, including *PAX1* [48], *KiSS1* [49], *CDKN2A*, *RASSF1* [50], *p14^AR^*, and *p16^INK4A^* [51], by localizing on methylated CpG islands and inducing hypermethylation and histone deacetylation via the recruitment of DNMT1 and HDAC1. Inversely, the downregulation of UHRF1 in MKN45 gastric cancer cells induced promoter demethylation and reduced cellular proliferation through the activation of different tumor suppressor genes [52]. Previous studies have also identified histone H3K9 methyltransferase (HMTase) G9a as an epigenetic regulator of UHRF1 [43]. Specifically, it was shown that the increased expression of G9a along with the transcription factor YY1 specifically repressed UHRF1 transcription during TPA-mediated leukemia cell differentiation [43].

In the context of ALL, UHRF1 was shown to be a negative regulator of the macrophage migration inhibitory factor (*MIF*)’s oncogene by binding to the CATT repeat sequence of the *MIF* promoter. Moreover, a UHRF1 knockdown in T-ALL cells resulted in *MIF* deficiency, with resultant apoptosis of T-ALL cells and significant improvements in the survival time of transplanted mice compared with that of the respective controls [53]. Consistent with these findings, we also showed that UHRF1 knockdown in T-ALL and B-ALL cells increased apoptosis-related proteins cleaved PARP and cleaved caspase-3.

Oncogenic c-Myc is reported to play an important role in promoting hematopoiesis and T-ALL expansion [54,55]. Interestingly, c-Myc was reported to be the upstream transcriptional regulator of UHRF1 in germinal center B cells [56]. In our study, we demonstrated that UHRF1 directly interacts with c-MYC and that the knockdown of UHRF1 in T-ALL and B-ALL cells reduces c-Myc protein expression. We posited that reduced c-Myc protein expression may be attributed to protein degradation and stabilization. To this end, a previous study demonstrated that UHRF1 regulates ROR1 protein expression indirectly by preventing ROR1-mediated ubiquitination in [1,27] pre-B-ALL and other malignancies [19,57]. Moreover, it was demonstrated that UHRF1 maintains the survival of a t(1;9)-positive pre-B ALL cell line in a ROR1-dependent manner [19]. The silencing of UHRF1 in t(1;19)-positive pre-B ALL cells significantly reduced their viability through the reduction of the ROR1 protein level [19]. Collectively, these findings are indicative of the multiple roles in which UHRF1 modulates protein expression.

## 5. Conclusions

Our study points to a strong association between UHRF1 and ALL, with UHRF1 as a regulator of c-Myc, CDK4/6, and Rb phosphorylation, and provides a rationale for the pursuit of UHRF1 as a potential therapeutic target in ALL.

## Figures and Tables

**Figure 1 cancers-14-04262-f001:**
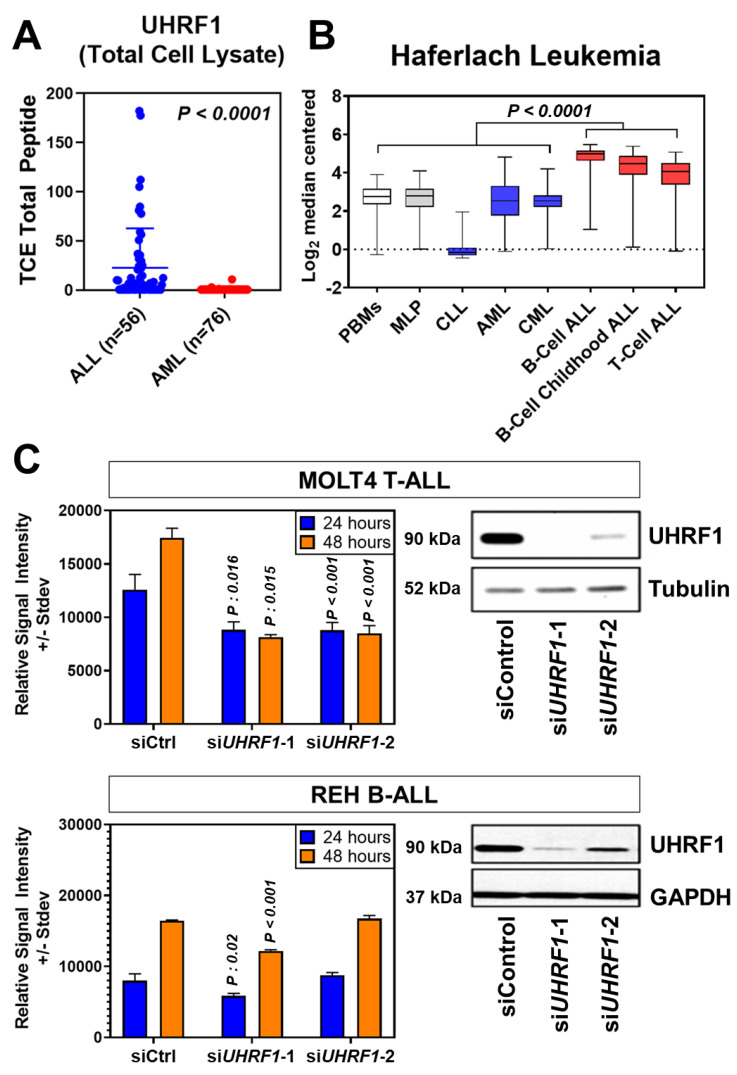
UHRF1 protein and gene expression in leukemia subtypes and their knockdown effects on cell proliferation. (**A**) Scatter plot of UHRF1 protein expression in patient-derived primary ALL (*n* = 56) and AML samples (*n* = 76). *p* = 0.0001 vs. AML samples based on mass spectrometry spectral counts of UHRF1 tryptic peptides. (**B**) Haferlach Leukemia Cohort transcriptomic dataset was used to evaluate the expression of UHRF1 in different subtypes of leukemia. (**C**) Knockdown efficiency of UHRF1 siRNA in MOLT4 (T-ALL) cells. siRNA knockdown efficiency was confirmed after 48 h by immunoblotting, and an MTS assay was performed at 24 and 48 h after transfection. Each experiment was performed in triplicate (*n*  =  3). The uncropped blots are shown in Appendix A.

**Figure 2 cancers-14-04262-f002:**
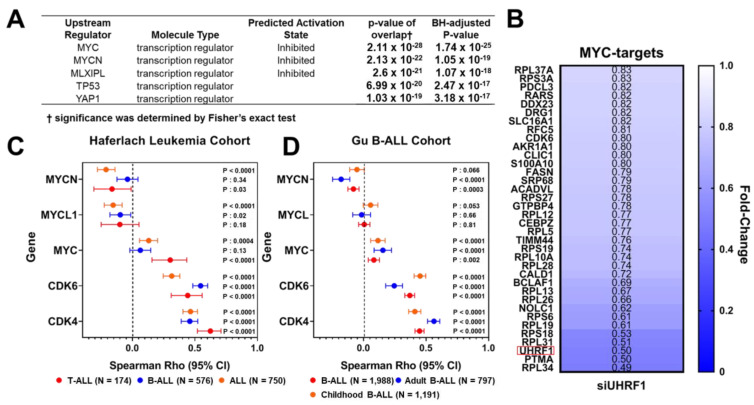
UHRF1 interacts with c-Myc protein. (**A**) An Ingenuity Pathway Analysis of 345 differentially expressed proteins (fold change ≥ 1.2 or ≤0.83) following a siRNA-mediated knockdown of UHRF1 in MOLT4 T-ALL cells compared with that of siControl. (**B**) Heatmap depicting fold-change in protein levels of annotated downstream MYC-targets in MOLT4 T-ALL cells following a siRNA-mediated knockdown of UHRF1 compared with that of siControl. Fold change < 1 indicates that the protein expression was reduced following a siRNA-mediated knockdown of *UHRF1*. Red box highlights UHRF1. (**C**,**D**) Distribution plot illustrating Spearman r coefficients (95% CI) for the association between gene expression of UHRF1 and mRNA levels of MYC, MYCL1, MYCN, CDK6, and CDK4 amongst the ALL and ALL subtypes in the Haferlach (**C**) and Gu (**D**) Leukemia Cohorts [29,30].

**Figure 3 cancers-14-04262-f003:**
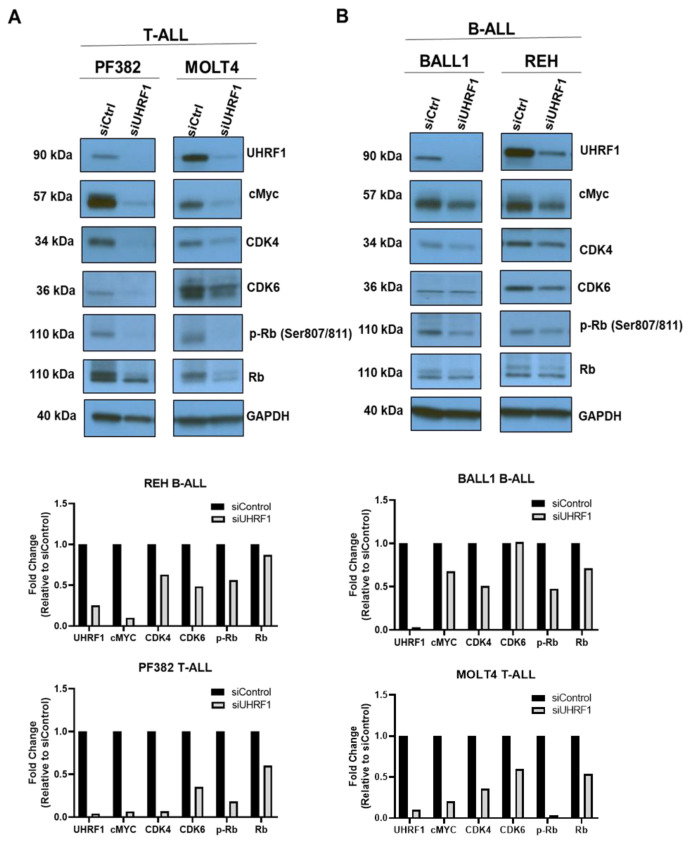
Regulation of the c-Myc-CDK4/6-phosphoRb axis by UHRF1 in B-ALL and T-ALL. A knockdown of UHRF1 led to reduced c-Myc protein in (**A**) T-ALL and (**B**) B-ALL cells. The ALL cells were transfected with siRNA-targeting UHRF1 and a control siRNA. After being incubated for 48 h, the UHRF1, c-Myc, CDK4/6, and p-RB/RB protein levels were analyzed using Western blot and densitometry. The uncropped blots are shown in Appendix A.

**Figure 4 cancers-14-04262-f004:**
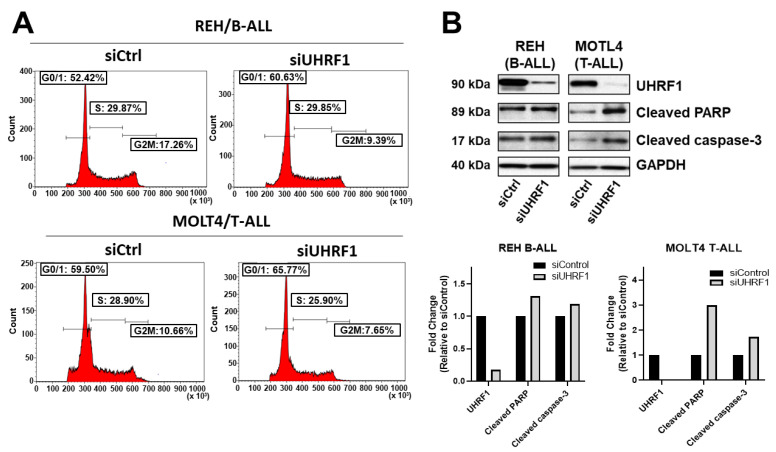
Knockdown of UHRF1 induces accumulation of cells in the G0/G1-phase and apoptosis. (**A**) Cell cycle analysis of REH B-ALL and MOLT4 T-ALL cells following a siRNA-mediated knockdown of UHRF1. (**B**) Immunoblots for UHRF1, cleaved PARP, and cleaved caspase-3 following a siRNA-mediated knockdown of UHRF1 in REH B-ALL and MOLT4 T-ALL cells. Densitometry measurements normalized against GAPDH loading control are provided. The uncropped blots are shown in Appendix A.

**Figure 5 cancers-14-04262-f005:**
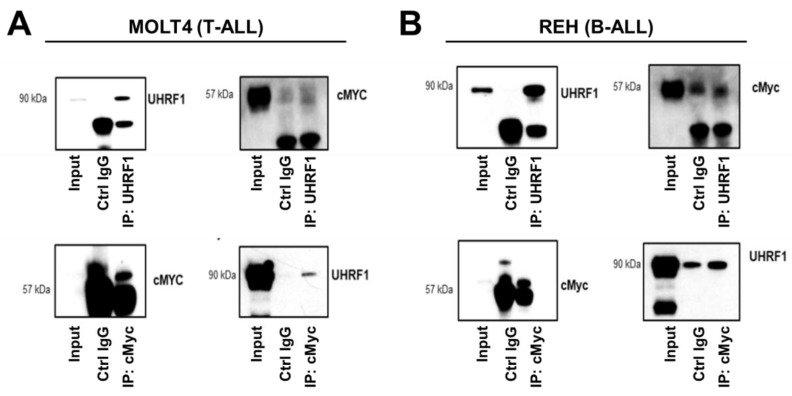
UHRF1 directly interacts with c-Myc. Co-immunoprecipitation/mass spectrometry was performed to determine the interaction between UHRF1 and c-MYC in (**A**) MOLT4 (T-ALL) and (**B**) REH (B-ALL). Lysates from MOLT4 and REH cells were subjected to co-immunoprecipitation with anti-UHRF1 and anti-c-Myc antibodies. The UHRF1 and c-Myc interacted with each other at endogenous protein levels. The uncropped blots are shown in Appendix A.

## Data Availability

The data presented in this study are available in this article (and Appendix A).

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
