# Peer review of "Novel UHRF1-MYC Axis in Acute Lymphoblastic Leukemia"

_cancers, 2022, doi:10.3390/cancers14174262_

Round 1

Reviewer 1 Report

Park et al have adequately addressed the concerns cited in my primary review. The authors have improved the manuscript.

Reviewer 2 Report

The authors have adequately addressed my concerns.

Reviewer 3 Report

The manuscript is acceptable in its revised version

This manuscript is a resubmission of an earlier submission. The following is a list of the peer review reports and author responses from that submission.

Round 1

Reviewer 1 Report

This is an interesting paper. It is generally well written. I have a few comments.

Nevertheless, the discussion section of this article could be benefit from a more elaborated debate. A more detailed explanation of the genetic and epigenetic role of UHFR1 is highly recommended. Furthermore, the discussion section would be improved by critically commenting on related publications in which the role of this gene has been silenced in ALL cell lines by using other methodologies such as CRISPR.

More evidence is required to conclude that UHRF1may be a therapeutic target.

Reviewer 2 Report

The authors identify Ubiquitin-like containing PHD and RING finger domain (UHRF) family member 1 (UHRF1) as an upstream regulator of MYC and proliferative signaling in B-ALL and T-ALL. Proteomic analysis revealed an upregulation of UHRF1 when compared to AML. Knockdown of UHRF1 in ALL cell lines resulted in decreased proliferation, reduced MYC protein expression and markedly reduced expression of MYC target genes, as identified in a proteomic screen and further validated by western blot. The authors use co-immunoprecipitation-mass spectrometry to show a direct interaction of UHRF1 and MYC in ALL cell lines, leading to the conclusion that UHRF1 stabilizes MYC in this context. These findings provide novel insights contributing to the complex picture of UHRF1 functions in cancer. The study is methodologically robust and very clearly described. However, some points should be addressed before considering this manuscript for publication.

MAJOR POINT:

1.) The authors point out that UHRF1 is overexpressed in B- and T-ALL. This is clearly shown for the comparison of protein expression to AML and less clear for the comparison of gene expression to CLL, AML and CML. What was the rationale to use a multi comparison test here? Was the individual comparison between entities also statistically significant?  UHRF1 seems to be high expressed also in normal lymphatic cells. To substantiate the conclusion that UHRF1 is overexpressed in ALL, its expression should be compared between ALL and normal lymphoid progenitors. Otherwise, it still seems possible that this upregulation represents a cell-type specific rather than a leukemogenic function.

MINOR POINTS:

1.) The authors conclude from co-IP experiments that UHFR1 stabilizes MYC. This conclusion is based on the direct interaction of both proteins. Either additional experimental data or a more specific backup from the literature seems warranted to support this conclusion, which otherwise should be indicated as hypothesis.

2.) The title of manuscripts links the newly established UHFR1-MYC axis to ALL disease progression. This wording has a very strong clinical connotation, while only pre-clinical data is presented regarding proliferative functions of this axis. The authors might want to consider using more specific wording.

Reviewer 3 Report

In this manuscript, the authors studied the expression of UHRF1 in leukemia and found that it is highly expressed, specifically in ALL. They demonstrated, using siRNA that knockdown of UHRF1 in MOLT and REH cells reduces viability and affects the expression of MYC and CDKs. They conclude that UHRF1 stabilizes MYC and suggest its potential as a therapeutic target in ALL.

I have the following comments:

  • In the heatmap shown in Figure 2B the fold change in the protein levels of MYC downstream targets following UHRF1 siRNA compared to control siRNA is not remarkable. Additionally, CDK4, whose protein levels are lower than CDK6 levels in MOLT 4 cells after UHRF1 siRNA treatment (Fig 3A) is not among the list of downregulated MYC targets shown in the heatmap in Fig 2B. I strongly suggest repeating this experiment in Fig 2B using a different cell line to confirm the results.

  • In Figure 3B, CDK6 levels are not affected in the B-ALL cell line BALL1 after UHRF1 siRNA, rather CDK4.

  • Although co-immunoprecipitations show that c-Myc and UHRF1 exist in the same complex, this doesn’t necessarily mean that UHRF1 stabilizes MYC. I suggest conducting Ubiquitination assays to confirm this notion.

  • It is clear from the Haferlach and GU datasets that there is an association between UHRF1 and MYC and its targets, however, more functional studies are required to provide insight of how MYC is regulated by UHRF1.

  • Analysis of the cell cycle phases after UHRF1 knockdown is necessary to confirm the effect of pRB reduced phosphorylation shown in Fig 3.

  • How does UHRF1 knockdown reduces viability or MOLT4 and REH cell lines (Fig 1)? It is recommended to conduct apoptosis assays in these cell lines after UHRF1 knockdown.

  • How does knockdown of MYC affect the levels of UHRF1? Have the authors conducted these experiments?